

# Observability of fine-scale ocean dynamics in the Northwest Mediterranean Sea

Rosemary Morrow[1], Alice Carret[1], Florence Birol[1], Fernando Nino[1], Guillaume Valladeau[2], Francois Boy[3], Celine Bachelier[4], Bruno Zakardjian[5]

[1]LEGOS/OMP/IRD/CNRS, Toulouse, 31400, France
[2]CLS Ramonville St-Agne, 31520, France
[3]CNES, Toulouse, 31400, France
[4]IRD, Brest, 29280, France
[5]Institut Méditerranéen d'Océanologie, La Garde Toulon, 83957, France

*Correspondence to*: Rosemary Morrow (Rosemary.Morrow@legos.obs-mip.fr)

**Abstract.** Technological advances in the recent satellite altimeter missions of Jason-2, Saral/AltiKa and Cryosat-2 have improved their signal-to-noise, allowing us to observe finer-scale ocean processes with along-track data. Here, we analyse the noise levels and observable ocean scales in the northwest Mediterranean Sea, using spectral analyses of along-track sea surface height from the three missions. Jason-2 has a higher mean noise level with strong seasonal variations, with higher noise in winter due to the rougher sea-state. Saral/AltiKa has the lowest noise, again with strong seasonal variations. Cryosat-2 is in SAR mode in the Mediterranean Sea but with lower resolution ocean corrections; its statistical noise level is moderate with little seasonal variation. These noise levels impact on the ocean scales we can observe. In winter, when the mixed layers are deepest and the sub-mesoscale is energetic, all of the altimeter missions can observe wavelengths down to 40-50 km (individual feature diameters of 20-25 km). In summer when the sub-mesoscales are weaker, Saral can detect ocean scales down to 35 km wavelength, whereas the higher noise from Jason-2 and Cryosat-2 blocks the observation of scales less than 50-55 km wavelength.

This statistical analysis is completed by individual case studies, where filtered along-track altimeter data are compared with collocated glider and HF radar data. The glider comparisons work well for larger ocean structures, but observation of the smaller, rapidly moving dynamics are difficult to collocate in space and time (gliders cover 200 km in a few days, altimetry in 30 secs). HF radar surface currents at Toulon measure the meandering Northern Current, and their good temporal sampling shows promising results in comparison to collocated Saral altimetric currents. Techniques to separate the geostrophic component from the wind-driven ageostrophic flow need further development in this coastal band.

## 1 Introduction

The ocean circulation in the northwest Mediterranean Sea exhibits widespread mesoscale dynamics, with strongest values along the Northern Current which flows westward along the French coast following the continental slope (Millot, 1999 ; Guilhou et al., 2013). Observing the mesoscale variability is critical in this region since it plays a key role in the coupled



ocean-atmospheric system that can lead to extreme precipitation events (Lebeaupin Brossier et al., 2015). Horizontal currents stirred by the mesoscales are important in the dispersion of pollutants and the monitoring of marine ecosystems. The vertical transport of heat, salt and nutrients is strongly driven by the smaller-scale dynamics in the fronts and filaments surrounding these mesoscale eddies, and within the deep convection cells that form in the Gulf of Lyons in winter/spring (Hermann et al., 2008).

Compared to other current systems at similar latitudes such as the Gulf Stream, the mesoscale variability in the Northwest Mediterranean Sea has a small Rossby radius of 5-15 km, varying seasonally with the stratification (Grilli and Pinardi, 1998). This makes the ocean dynamics of this region particularly difficult to observe and monitor. The surface mesoscale characteristics have been studied with satellite SST and ocean colour data in clear-sky conditions (Robinson, 2010), but the mesoscale variability is often hidden in winter by clouds and in summer under the more homogenous warm surface layer. Numerical modelling studies are improving in resolution and in their internal physics to allow a better representation of the mesoscale variability (eg, Hu et al., 2016; Hermann et al., 2008), although these models need to be validated against observations.

In the global ocean, mapped satellite altimeter products have allowed unprecedented advances in understanding the mesoscale eddy variability and characteristics (Chelton et al., 2011). Altimetry measures sea surface height (SSH) that responds to mass and density changes over the entire water column, and as such, altimetry is the only satellite observation that can detect deep ocean changes. Deep-reaching mesoscale eddies can be tracked over many seasons or years (eg Morrow et al., 2008; Chelton et al., 2011), even if their surface signature disappears through air-sea interactions, and become undetectable in satellite imagery. Although regional altimeter maps have been constructed with improved resolution and spatial scales adapted for the Mediterranean Sea (e.g., Pujol and Larnicol, 2005), the spacing between groundtracks still limits our ability to monitor scales less than 150 km wavelength (or 75 km diameter features) (Pascual et al., 2006). Thus we can only detect the larger mesoscale structures, missing most of the typical Rossby radius dynamics in the Mediterranean Sea.

Along-track altimeter data are able to detect finer scales than the mapped altimeter data, but the spatial scales we can resolve are still limited by the altimeter noise, the accuracy of the corrections and the processing methodology. However, over the last 5 years, there has been great progress in improving the quality of along-track satellite altimeter data for ocean studies. Of the 3 missions currently flying in the altimeter constellation, Jason-2 in Ku-band (launched in 2008) has benefitted from continually refined algorithms and corrections, and new waveform retrackers that allow more data points to be collected close to the coast and islands, and more stable performance with lower noise over the oceans (Dibarboure et al., 2011). Saral/AltiKa (launched in 2013) was designed to have a smaller footprint and lower noise over all surfaces, due to the choice of antenna pattern, Ka-band frequency and its lower altitude (Verron et al., 2015). Cryosat-2 (launched in 2010) is primarily a cryosphere mission and not planned for ocean observations. Yet over the last years, considerable efforts have been made by the ESA SAMOSA project (Ray et al., 2015) and the CNES CPP project (Boy et al., 2012) in collaboration with oceanographers to improve the waveform retracking over the ocean and provide adequate corrections for ocean observations.



Cryosat-2 is in low-resolution mode over most of the global ocean but has Synthetic Aperture Radar (SAR) mode observations available over a few regions, including the Mediterranean Sea, with improved along-track sampling down to 300 m and reduced noise. However, certain ocean corrections are less accurate than on Jason-2 or Saral, including the radiometer correction and the mean sea surface estimate since Cryosat-2 is on a geodetic orbit. These three altimeter missions with different technologies and data processing will provide an ideal data set to test the improved observational capabilities in the NW Mediterranean Sea.

Previous studies have analysed the altimetric capabilities in the NW Mediterranean Sea from conventional along-track data (Bouffard et al. 2008; 2011; Birol and Delebeque 2014; Birol and Nino, 2015)), including using seasonal averaging to reduce the noise for Jason but maintaining along-track resolution (Birol et al., 2010). Here we will take a different approach, in order to measure the altimetric signal-to-noise statistically in the different seasons,. We will calculate along-track SSH spectra (eg Fu, 1983), which allows us to observe the SSH spectral energy at different wavelengths, and also the time-averaged spectral noise at small wavelengths. In terms of signal, the spectral energy of SSH is higher at longer wavelengths, and lower at small wavelengths, and geostrophic turbulence theory involves a cascade of energy from the larger to smaller scales, leading to a steep spectral slope in wavenumber space. When spectra are averaged (over different groundtracks in a region and/or over time along the same groundtrack), the random altimeter noise averages out to create a flat spectral noise floor in the 1 Hz data. This spectral noise level then defines our altimeter noise. The intersection of this noise floor with the spectral slope will define the limit of the observable wavelengths, where the signal-to-noise is statistically greater than 1.

Following Xu and Fu (2012) we will remove the spectral noise from the spectra before calculating the spectral slope, to improve the slope estimate and have more precise observational limits. This technique has been applied to the global altimeter data sets, for Jason-1 by Xu and Fu (2012) and for Jason-2, Saral and Cryosat-2 by Dufau et al. (2016). Their results showed considerable geographical variations in the spectral slope, noise levels and mesoscale resolution (Xu and Fu, 2012), and strong seasonal variations in the noise level and the mesoscale observing capabilities (Dufau et al., 2016). Neither study included the smaller Mediterranean Sea region, due to the limited spatial coverage in this regional sea. In our analysis, we will concentrate on tracks having at least 200 km length.

The Mediterranean Sea, dominated by small dynamical structures, may have different spectral energy and spectral slopes than in other open ocean regions. The surface sea-state conditions are also dominated by short wind-waves and less by long swell, which may impact on the radar altimeter's noise level. Both of these features will be considered in the first section of this paper. We aim to investigate the noise levels for the most recent altimeter missions, estimated from their spectral noise level in the Mediterranean Sea. We will revisit the appropriate filtering to be applied to remove the noise in different seasons. We will then consider what scales of ocean dynamics can be observed today in the Mediterranean Sea with along-track altimetry, and investigate how much of the seasonal dynamical signal is observable above the seasonal noise.

In the second part of this paper, we will use a complementary approach, and focus on the observation of individual features using a combination of altimetry and a limited number of glider sections and two years of HF radar observations filtered at similar scales. We will examine whether the ocean scales observable with altimetry are also captured by the collocated in-



situ data. Glider-altimetry comparisons have been used for previous altimetry missions in the NW Mediterranean Sea (eg Bouffard et al., 2010), but not for the three most recent missions. For the glider comparison, we only have a limited number of historical collocated sections, and so gliders were deployed specifically along altimetric tracks for each of the three missions, under different mesoscale conditions. For the HF radar, we will use a HF radar site near Toulon, as part of the MOOSE observational array (Quentin et al., 2013), with an offshore extent of 25-75 km from the coast. We will discuss the strengths and limits of the different measurement systems' observation in the coastal band.

## 2 Data Sets Used

### 2.1 Altimeter data

Along-track SSH observations from the most recent altimetry missions (Jason-2, Cryosat-2 and SARAL/Altika) are analyzed over the NW Mediterranean Sea (Fig. 1) and over different periods (Table 1). The data are made available from AVISO/CNES. Jason-2 is a conventional pulse-width limited altimeter operating in Ku-band (Lambin et al., 2010), and provides the longest time series: we use data over the 6.8-year period from July 2008 to Feb 2015. SARAL/Altika with its 40 Hz Ka-band emitting frequency, its wider bandwidth, lower orbit, increased Pulse Repetitivity Frequency and reduced antenna beamwidth, provides a smaller footprint and lower noise than the Ku-band altimeters (Verron et al., 2015). We use data from the near 2-year period from March 2013 to January 2015. Cryosat-2 is a Siral Ku-band instrument operating in three modes (Low-Resolution Mode LRM, Synthetic Aperture Radar Mode (SARM) and SAR interferometric mode). Only the SARM data are available over the Mediterranean Sea, and we use data from the CNES Cryosat-2 processing prototype (version 14) from CNES (Boy *et al*., 2012) over the one-year period April 2013 to April 2014. For all 3 missions we will analyse the 1 Hz data only which has a flat noise floor – the higher frequency (20 Hz or 40 Hz data) show a spectral bump at wavelengths less than 70 km which does not allow us to estimate a stable noise floor (Dibarboure et al., 2011).

The choice to analyse different periods was dictated by the data availability, and our desire to have longest possible time periods available for the seasonal analyses. The limited quantity of altimeter cycles considered during this period is compensated by the spatial averaging of available tracks in the NW Mediterranean Sea that improves the statistical significance of our analysis.

Along-track SSH observations are maintained at their original observational position, and corrected for all instrumental, environmental and geophysical corrections. Only the time variable part of the SSH is considered following Stammer (1997), Le Traon *et al*. (2008) and Xu and Fu (2011, 2012). Sea level anomalies are calculated for all missions relative to their precise along-track mean sea surface for Jason-2 and Saral, both on a long-term repeat track. Cryosat is on a geodetic orbit, and its sea level anomalies are calculated relative to a gridded mean sea surface (MSS_CLS2011, http://www.aviso.altimetry.fr/en/data/products/auxiliary-products/mss.html), which can introduce slightly higher errors over scales of 40-80 km wavelength (Dibarboure et al., 2011; Dufau et al., 2016).



## 2.2 Glider data

A large number of gliders have been deployed in the NW Mediterranean Sea as part of the MOOSE project (http://www.moose-network.fr/gliders), with more than a hundred glider sections available in the region during the 6.5 years of our study. However, since our objective was to validate the smaller scale structures that move rapidly, it was important

that the glider and altimeter observations were collocated in space and time. Two glider sections were available along a Jason-2 track from Sept-Oct 2012. MOOSE and CNES also co-funded the deployment of gliders along 3 Saral tracks as part of the Comsom campaign in Oct-Nov 2014, and along 2 Cryosat-2 tracks and 3 Saral tracks in April-May 2015, see Fig. 5a and Table 2.

Slocum gliders were used, diving at a 26° inclination with an average horizontal speed of around 0.35 m/s. They reach a

maximum depth of 1000 m, and the distance between two surface positions is around 2-3 km. The deployments are made away from the coast to be in deep water, although an onboard captor can detect if they approach the bottom before 980 m. The gliders were deployed a few days before the passage of the satellite in order to be sampling along the track when the altimeter passed. The altimeter passes every 10 days for Jason, and every 35 days for Saral and in a given region every month for Cryosat-2. So with this type of precise-date deployment, there is no guarantee that the glider and altimeter pass

will cross an energetic structure at the time and position that the altimeter passes.

For comparison with the altimeter data, we need to obtain steric heights from the glider relative to 1000 m. For this, we calculate a single vertical profile at the central position for each of the diagonal dives (descending or ascending), and calculate steric heights from the density anomalies. Geostrophic velocities are also calculated relative to the 1000 m depth.

There is an additional "drift" speed that can be added to this geostrophic velocity, associated with the lateral heading

correction used to keep the glider on track against a strong current. This drift correction represents the total current over the upper 1000 m and will include the barotropic currents close to the continental slope, some ageostrophic surface currents and a correction for the upper baroclinic flow. This correction was generally small in our region except near the continental slope, and we will clearly identify when this correction is used in the following study.

## 2.3 High-Frequency (HF) Radar data

As part of the MOOSE observing system, a HF radar system has been installed near Toulon over a number of years (http://hfradar.univ-tln.fr/HFRADAR), to monitor the Northern Current. The system is composed of two HF radars with a lateral incidence angle that measure the reflected radar signal from the ocean surface. By measuring the frequency of the energy peak from the waves and knowing the frequency of the emitted radar signal, one can estimate the surface movement. The surface currents are then obtained after subtracting the surface wave speed. Two radars orientated with different angles

allow the determination of the current direction.

The system uses two WERA radars located to give surface current vectors over a region extending 80-100 km offshore, with a spatial resolution of 3 km and an angular resolution of 2°, and operating at 16-17 Mhz. Observations are made every 20



mins and the data are edited and averaged daily over the period May 2012-Sept 2014. The surface current vectors represent the total current averaged over the upper 1 m of the ocean, and include a significant ageostrophic component, not present in the altimetric currents.

## 3 Spectral Analysis of along-track altimeter data

Spectral analyses are performed on each of the three altimeter missions, with their tracks shown in Fig. 1. Only data more than 50 km from the coast are analysed to remove the increased errors in the coastal zone. Each track and cycle is then selected along a common segment of 200 km. Missing data are a problem for a stable spectral analysis. If less than 3 consecutive 1 Hz points are missing (20 km) the data are linearly interpolated, if a larger gap is present the cycle is eliminated from the analysis. Tracks passing over large islands are thus eliminated (See Fig. 1). Wavenumber spectral

analysis is then performed by Fourier transform on the ensemble of the remaining segments for each mission (see Table 1). The cycles are averaged in wavenumber space for the entire period, and for each season. An example of the power spectral density (PSD) of sea level anomaly averaged for all of the Jason-2 data in the NW Mediterranean Sea is shown as the black curve in Fig. 2. The PSD is higher at longer wavelengths (> 300 km), there is a cascade of energy over the mesoscale range from 50-300 km, but the spectra becomes whiter at small wavelengths (i.e., less than 50 km), where the weaker ocean energy

is hidden by the stronger instrument and geophysical noise.

This *noise level* is then calculated as a constant PSD value estimated between 12 and 25 km wavelength, as in Dufau et al. (2016) (red line, Fig. 2).

Following the global studies made by Xu and Fu (2012) and Dufau et al. (2016), we then subtract this statistically stable noise level from the mean spectral curve, to obtain an unbiased spectral estimate corrected for the noise (thick black curve,

Fig. 2). The *spectral slope of this unbiased estimate* is steeper over the mesoscale range and corresponds to a $k^{-2.5}$ slope and the SSH PSD cascade continues more smoothly down to smaller wavelengths.

We define the *mesoscale observability limit* as the wavelength corresponding to the intersection of the spectral slope and the noise level, where the signal-to-noise ratio is greater than 1. This is a statistical representation of the average ocean and noise conditions over the entire period and over the entire region analysed. In some local cases, smaller energetic structures may

still be observable above the altimetric noise. However in the following results, we will discuss this regional statistical approach.

The *mean spectra* for the 3 altimeter missions over the NW Mediterranean Sea are shown in Fig. 3a for the 200 km segment tracks in Fig. 1, and over the 13-month common data period from 1 April 2013 to 30 April 2014. The unbiased estimate with the noise removed is in Fig. 3b. Recall that the space-time sampling of the 3 missions are different, and as such they may

capture different dynamics at different regions. So we don't expect the spectra to be perfectly aligned. More distinctive is the different noise levels between 15-100 km wavelength. Jason-2 has the highest noise level in this region, followed by Cryosat-2 in SAR mode. Saral/AltiKa in Ka-band exhibits the lowest noise of all.



When a constant noise level is removed from each spectral PSD, the spectral slopes line up surprisingly well, given the different space-time sampling of the 3 missions over this 13 month period. The spectral slope is again around $k^{-2.5}$ from a fit to the unbiased spectra over the wavelength range from 50-200 km. These spectral slopes in the offshore regions of the Mediterranean Sea are quite shallow compared to the $k^{-5}$ slopes expected for quasi-geostrophic theory (Stammer, 1997). The reason for this needs further investigation, but smaller slopes are also characteristic of open ocean low eddy energy regions (Xu and Fu, 2012). For the Mediterranean Sea, the dominant mesoscale energy at small Rossby radius scales tends to flatten the spectra, but internal waves or mean sea surface errors in the Cryosat-2 data could also contribute to higher SSH energy at small scales, and flatter spectra (Dufau et al., 2016).

The fact that the Cryosat-2 1 Hz data in SAR mode had a higher noise level than Saral/AltiKa was unexpected. We verified that the Cryosat-2 20 Hz data were consistent with the 1 Hz averages, so this is not an averaging problem. The Cryosat-2 20 Hz SAR mode does exhibit a spectral hump for this region and time period that was not present in other regions with SAR data (Agulhas or Tropical Pacific; *S. Labroue, personal comm.*). This warrants further analysis of the particular surface roughness conditions occurring in the NW Mediterranean during this year, and further expertise in SAR processing for the Mediterranean conditions is needed. These results reinforce the very low noise level associated with the 40 Hz Ka-band SARAL data, averaged here to 1 Hz.

Seasonal spectra were also calculated from the longest time series possible, ie over 6.5 years for Jason-2 data, over 22 months for Saral/AltiKa, and for the shorter 13 month period or Cryosat-2 (see Table 1). The spectral noise floor levels for the seasonal analyses are shown in Fig. 4a. Jason-2 and Saral/AltiKa show a large seasonal variability in their noise levels, with highest noise levels in winter (12 cm$^2$/cpkm) and then autumn, due to the high sea-state roughness in these months from the stronger wind-waves conditions which increases the spectral SSH "hump" at wavelengths from 30-70 km (Dibarboure et al., 2014). In summer, the Jason-2 noise level is only 8 cm$^2$/cpkm, but this is still higher than the noise floor in any season for the Saral or Cryosat-2 missions. Saral with its small footprint has the lowest noise levels but has strong seasonal variability, with values ranging from a low 3 cm$^2$/cpkm in summer to 7 cm$^2$/cpkm in winter. The Cryosat-2 SAR mode shows very stable background noise levels over this one year record, varying between 6 to 8 cm$^2$/cpkm. The reasons for this stable seasonal noise level are not yet known. However Cryosat-2 has a long repeat cycle (369 days), so different geographical regions are sampled in different seasons, there may be strong interannual variations in the wind-wave conditions that merit more detailed investigation. The additional MSSH errors introduced due to the non-repeating track will also impact the Cryosat-2 spectra over all seasons.

Figure 4b shows the observational limits for each altimeter mission by season. Clearly, the background noise is not the only limiting factor on the scales of mesoscale energy that we can observe. The SSH energy at low wavelengths also varies from one season to another. In winter, when the mixed layers are deepest and energetic deep convection cells occur in the NW Mediterranean Sea (e.g., Hermann et al., 2008), all of the altimeter missions can observe wavelengths down to 40-50 km (individual features of 20-25 km). In summer when the sub-mesoscales are weaker, Saral can detect ocean scales down to 35 km wavelength, whereas the higher noise from Jason-2 and Cryosat-2 blocks the observation of scales less than 50-55 km.





This characteristic was also found in global analyses by Dufau et al. (2016). Unfortunately in winter, when we would like to observe the smaller energetic submesoscales, all of the radar altimeters observe higher noise levels associated with the higher wind-wave field.

## 4 Colocated altimeter and glider observations

The previous section highlighted that the altimetric noise was effectively masking the smaller-scale SSH signals in the along-track data. The smallest scales observable with a signal-to-noise greater than 1 will vary from one altimeter mission to another and seasonally. Statistically, we cannot observe structures less than 35-45 km wavelength with Saral, or 50-60 km wavelength with the higher noise of Jason-2. However, individual energetic features may be revealed above the statistical noise. We will explore this with a series of collocated along-track altimeter-glider sections, and compare the vertical

structure observed by the gliders with their steric height and geostrophic velocities.

One should bear in mind that the glider steric height and geostrophic velocities (with or without their surface drift adjustment) will observe different dynamics from the altimetric sea level and geostrophic velocity anomalies. The steric height calculated from gliders represents the upper ocean baroclinic component due to the density anomalies above 1000 m depth. Altimetric sea level anomalies include the full-depth baroclinic motions and the barotropic component, and the

barotropic flow may be quite active in the NW Mediterranean Sea, in particular near the shelf break and slope (F. Lyard, pers communication). When the glider "surface drift" is added to the glider geostrophic currents relative to 1000 m, this may partially correct for the missing barotropic component. Altimetry may also include other SSH signals, such as from internal tides or internal waves, which contribute as errors in the geostrophic velocity calculation (although tides are small in the Mediterranean Sea). In addition, the altimetric sea level anomalies have the mean ocean circulation removed, whereas the

gliders provide the total upper ocean baroclinic flow. For consistency, the mean dynamic topography and mean geostrophic velocities derived from Rio et al. (2014) are added to the altimetric data for this comparison. The third main difference is the time taken to make a section over 100 to 300 km. The altimeter makes a "snapshot" of the section as it passes at 7 km/sec (200 km in 30 secs) whereas the glider moves at 0.35m/sec (200 km in 6.5 days). We will see that slow-moving structures may be well-sampled by both; rapidly-evolving smaller-scale structures are harder to collocate.

One crucial point is that the gliders have their own noise and also measure high-frequency ageostrophic ocean structures that will not be observable with altimetry. Figure 5 shows a vertical temperature section over the upper 200 m from the glider "Milou" along the Saral altimeter track 57 from the 27 Oct to 3 Nov 2014. Figure 5b shows the very small-scale signals in the upper ocean temperature structure along this 164 km long section. These may be associated with noise in the glider heading or from the processing steps, or from internal waves or rapid sub-mesoscale structures. To remove these scales, we

have applied a recursive Butterworth order 2 along-track filter to the density data, before calculating the steric height or geostrophic anomalies, with a filter cut-off at 30 km wavelength, designed to retain the typical Rossby radius scales of 10-15



km in the NW Mediterranean Sea. An example of the filter applied to the same temperature section is shown in Fig. 5c. Similar filtering is applied to the different glider sections presented below.

Ten glider sections are available, collocated with altimeter tracks (details given in Table 2). Here we present three glider tracks sections along different altimeter mission tracks.

## 4.1 Jason-2 / Glider comparison over a large slow eddy

The glider "Campe" followed a Jason-2 track 146 over a 300 km section from 42°N to 39.5°N over a one-month period 23 Sept – 23 Oct 2012. During this period, Jason-2 passed 3 times over the same track. Jason-2 data were filtered using a Loess filter with a 50 km cutoff for this summer-autumn section (see section 3). Figure 6a shows the glider cross-track geostrophic currents (in red) with the Jason-2 cross-track currents superimposed (black) for the southward passage on 1 Oct 2012, overlaid on the satellite SST for the same date. The northward passage centred on 21 Oct 2012 is in Fig. 6c. The southbound section in late September has weak currents and is located slightly to the west; the northbound section crosses a strong mesoscale structure with an eastward current from 40.3-41.3°N, then a westward return current from 41.3 to 42°N at the northern end, when the 3$^{rd}$ Jason pass is collocated. The filtered glider data and the filtered Jason data are also shown for the southbound section (Fig. 6b) and the northbound section (6d). The instant of the Jason-2 passages is marked by a vertical line – identifying the latitude where the glider and the Jason observations exactly coincide in time. The geostrophic currents from the AVISO 2D maps are also shown for reference.

The southbound section crosses a series of small reversing currents around small SST structures of 30-50 km (Fig. 6a). The glider and along-track Jason-2 data show cross-track currents in phase, although the Jason-2 amplitudes are stronger (correlation, r=0.5; RMSE = 0.06 m/s). This may real (due to deeper baroclinic or barotropic structures not observed by the glider's upper 1 km observations), or induced by the effects of filtering higher noise. The mapped AVISO data have similar amplitude to the glider data, but are not in phase which reduced their statistical correlation (r=0.4; RMSE = 0.06 m/s). Adding the glider "drift" reference currents introduces little change to these results.

Three weeks later, the northbound section crosses a strong mesoscale eddy. The three data sets present similar eastward currents across the mesoscale eddy, and although the amplitude of the westward current near 42°N is similar, along-track altimetry positions the return flow 30 km further north than is detected by the glider. For this larger eddy, 100 km in diameter, the AVISO 2D maps and the 50 km filtered along-track data are both providing a good estimate of the glider's geostrophic currents (r=0.9) with similar RMSE (~0.07 m/s for both data sets).

## 4.2 Saral / Glider comparison over a small rapid meander

Although a number of satellite underpasses were planned for Saral, different deployment problems limited the number of successful intercomparisons (bad weather, gliders leaking, errors in estimating the satellite position, etc). The longer sections did not necessarily cross any energetic features, and we eliminated sections where the currents were always less than a few





cm/s. The short section presented here highlights another difficulty – comparing small-scale structures in a rapidly evolving field.

Figure 7a shows an example of the Saral – glider comparison for the Saral track 388 and the glider "Milou" which crossed a narrow intense westward current around 42.75°N, a broad, weak, westward current further south, then touched an eastward
return flow around 42.25°N. These narrow currents are the limit of the observability with the gliders, given the filtering cutoff at 30 km wavelength. In comparison, the altimeter data shows a broad intense westward flow over the entire section, except for the return eastward flow in the south. The along-track comparison of their amplitudes (Fig. 7b) shows that the two systems are measuring similar currents at the exact time of the Saral passage (vertical line), but otherwise the broad intense westward flow captured by altimetry is not observed by the gliders. The mapped AVISO data is halfway between.

If the glider and altimeter observations are overlaid on a daily time series of Satellite SST maps, the differences between these two observations becomes clearer. Figure 8 shows the 5 days it took the glider to complete this 77 km section to 1000 m depth, and the evolving SST conditions during this period. On the 9 Nov 2014, the glider was in the south and crossed a cold eastward moving filament. On the 10 Nov, the glider is in weaker conditions. On the 11 Nov, the warmer westward flowing current starts to shift southward and on the 12 Nov, when Jason-2 passed over, the warm branch has extended south
to 42.3°N.

This example highlights the difficulty in comparing sections constructed from 5 days of glider data with the near instantaneous coverage from the along-track altimetry data. These small scale structures less than 50 km evolve quickly, and having observations that are not exactly collocated in space and time leads to large differences.

### 4.3 Cryosat-2 / Glider comparisons

The third example concerns two gliders deployed at 1-day intervals along the Cryosat-2 track 493, which passed on the 27 April 2015. Cryosat-2 SAR data are filtered at 35 km (see section 3). Figure 9 shows that the two gliders and the Cryosat-2 data detect well the westward flowing Northern Current near 42.5°N as well as an eastward return flow around 41.5°N. In contrast, the Cryosat-2 data overlay a weak cyclonic eddy centred on 42°N, which is also apparent in the mapped AVISO data, but is not detected by the gliders. The Cryosat-2 data are included in the AVISO maps, so the two products show
consistent results, though AVISO is smoother.

The along-track geostrophic currents (Fig. 9b) show that the two gliders, separated by one day, are observing the same features. However, the peaks in westward flow, detected by the gliders at 42.6°N and 42.1°N are slightly more intense with the Cryosat-2 observations and have shifted southward when the altimeter observed them a few days later. Tintin is one day in advance of Bonpland as they move southward, and the southward shift in the westward flow is also observed between
Tintin and Bonpland at 42°N. There is a good alignment of the eastward currents between the 3 observing systems around 41.7°N.

In summary, the glider-altimeter comparisons reveal the difficulty in validating the along-track altimetry data with observations that are not exactly collocated in time and space. The relatively slow gliders are able to capture the slower





moving larger eddies, as seen in our example with Jason-2, and highlighted by previous studies (Bouffard et al., 2010). However, the real improvement in altimetric signal-to-noise levels expected with Saral and Cryosat-2 were not revealed in these glider comparisons, mainly because at the time of these altimeter observations, rather weak signals were detected or the small-scale meanders were moving rapidly. In these cases, our observations are approaching the error levels of the two

systems. Small offsets in the structure of the Northern Current could also be introduced by the removal of a Mean Sea Surface from the Cryosat-2 data sets, which could induce cm/s errors on these small space scales (up to 80 km wavelength, Dufau et al., 2016). Although gliders can observe energetic small-scale structures in dedicated campaigns in the Mediterranean Sea (e.g., Bosse et al., 2015), the chance is small that these occur at the precise position and time when the gliders and altimeter tracks coincide. This comparison highlights the difficulty in setting up a validation campaign for

altimetric observations of small-scale rapidly moving dynamics.

## 5 Colocated HF Radar & Saral altimeter

HF radar provides an additional means to observe the oceanic surface currents. In comparison to the geostrophic component of the flow obtained with altimetry and gliders, HF radars measure the total surface current, due to balanced geostrophic and unbalanced ageostrophic currents (wind-driven, inertial, tidal currents, …). The daily data set we used has been processed to

remove the high-frequency tides and inertial currents, leaving the geostrophic and wind-driven currents. Figure 10 shows an example of the HF radar total currents for one date, the 20 Oct 2013 near Toulon, with the two coastal radar locations marked. The presence of the strong Northern Current is clearly visible in the 2D HF radar current vectors, with a central jet only 10 km wide, the current spanning 20 km to its edges. This is clearly below the statistical observability limits from the spectral analysis of the three altimeter missions. The offshore extent of the HF radar data is from 25-75 km from the coast,

which extends into the coastal band that was excluded from our spectral analysis, as having frequently "noisy" altimeter data and corrections. The small spatial coverage of the HF radar means that no Jason-2 data cross this region, although we have one Saral track passing through the centre (Fig. 10), and a number of non-repeating Cryosat-2 tracks. The angle of the Saral track shown in Fig. 10 is such that the cross-track geostrophic currents are mainly orientated in the principal direction of the Northern Current. For this date (20 Oct 2013), the amplitude of the HFradar currents, projected in the altimetric cross-track

direction (in red), is similar to the Saral cross-track currents (in black), reaching 70-80 cm/s within the Northern Current. Further offshore, the HFradar currents decrease gradually whereas the geostrophic altimetric currents are much weaker outside of the jet. The presence of agesotrophic currents in the HFradar data could contribute to this difference. Our statistical estimate of the spatial observability of Saral observations in autumn is around 35 km wavelength (section 3), representing feature structures across the current of around 17 km. Clearly at these scales, the 20-km wide Northern Current

can be observed by the Saral altimeter.

The advantage of the HF radar data set is its daily 2D coverage at fine resolution, so we shouldn't have the space-time offsets in the sampling of small-scale features that plagued the glider-altimeter comparisons. The disadvantage is that



altimeter data in the last 10-50 km from the coast are noisy, and the ageostrophic wind-driven component of the HF radar surface currents can be strong here, in the region with strong Mistral winds.

We have compared the observability of these near-shore currents with the finer resolution Saral altimeter time series, filtered at 35 km (see section 3). Saral data are available along this track every 35 days, and Fig. 11 shows the 18-month time series of cross-track surface velocities from the HF radar. The upper panel shows the full time series of HF radar currents projected perpendicular to the altimeter track, the middle panel shows the HF radar currents sampled at the same dates as the Saral altimeter passes, and spatially sampled at 7 km as for the 1 Hz altimeter data. The bottom panel shows the Saral 1 Hz geostrophic currents (mean and anomalies), filtered at 35 km. Saral clearly detects more of the offshore return flow than the HF radar can but is covering a similar data range as the HF radar to the coast. Along-track correlations of the HF radar and altimetric currents for this cross-track velocity component are between 0.7-0.9 for these 16 tracks, except for 4 dates, where the correlations drop below 0.5. The RMSE between the cross-track HF radar current amplitudes and the Saral current amplitudes is shown in Fig. 12. Dates with low correlations (<0.5) are marked with the vertical dashed line, and these have a higher RMSE. The RMSE is generally lower in the summer months when the wind is lower, and increases in winter.

Wind forcing of the ageostrophic currents may explain part of the difference. If we consider the daily time series of HF radar data (Fig. 11a) and extract the outliers in cross-track velocity $> 1\sigma$ standard deviation from the mean, these outliers are correlated at 0.84 with the cross-track wind at the same date (not shown). For the dates with low correlations – wind may play a role for one date (Dec 2013) but the other dates have relatively low wind. The differences with Saral are often associated with 10 km wide structures and close to the coast. This could be due to errors in either measurement system (e.g., for Saral: the nearshore wave height bias, wet tropospheric corrections, mean sea surface errors), but also from rapid events that are detected by the altimeter 8-sec "snapshot" but viewed differently with the HF radar one day averages (rapid meander, internal waves, etc). Future analysis of the higher-frequency HF radar data and the altimetric 40 Hz altimeter data, with appropriate filtering, may help elucidate some of these differences.

# 6 Discussion

The along-track altimeter spectral analysis allows us to estimate the mean dynamical scales that can be observed today with different altimeter technology and associated processing, and in different seasons. In winter, when the mixed layers are deepest and the sub-mesoscale is energetic, all of the altimeter missions can observe wavelengths down to 40-50 km (individual feature diameters of 20-25 km). In summer when the sub-mesoscales are weaker, Saral can detect ocean scales down to 35 km wavelength, whereas the higher noise from Jason-2 and Cryosat-2 blocks the observation of scales less than 50-55 km wavelength.

This is a statistical view. There are limits in applying this too assiduously, especially as these statistics are calculated from relatively short records for Saral, and only 13 months of reprocessed SAR data for Cryosat-2. We chose to analyse the longest time series possible for the seasonal calculations since the records are relatively short, however entire years should



be analysed to remove any sampling biases in these statistics. Given the long repeat time for Cryosat-2 we are also measuring different geographical regions in each season which can introduce biases in our basin-scale averages. Interannual variations also occur in the dynamics in response to interannual atmospheric changes, which can lead to different deep convection events from one season to another (Adloff et al., 2015). Analysing a longer time series of Saral and Cryosat data should improve the significance of these early results.

One application of this type of analysis is to improve the altimetric data post-processing to be adapted to the regional conditions. Today, along-track filtering is applied in a similar way to all altimeter missions to reduce the instrument and geophysical noise. Since consecutive altimeter points are laid down spatially, data are filtered spatially along the track to reduce this noise. Standard filtering in the AVISO along-track products DT2010 ranges from 55 km wavelength at high latitudes to around 250 km in the tropics (Dibarboure et al., 2011). The new AVISO products DT2014 apply lower along-track smoothing at 65 km wavelength, globally and for all missions (Pujol et al., 2016). This study suggests that the along-track filtering may be tuned in a regional study to be better adapted to the local dynamics and noise conditions. Thus in the NW Mediterranean Sea, filtering of Jason-2 data could vary seasonally from 50 km in winter to 60 km in autumn and spring (or a conservative 60 km year-round). Saral could have a smaller along-track filtering applied, to retain wavelengths greater than 35 km in summer-autumn and 45 km in winter. A filter cutoff of 50 km year-round could be suitable for Cryosat-2. Knowing how this statistical signal-to-noise ratio varies from one mission to another, and seasonally, is very useful for regional applications, for local process studies or for data assimilation.

The in-situ validation remains very limited in space and time, and did not allow us to confirm whether these smaller scales are realistic ocean features. For the glider comparison with SARAL, small scale structures were detected by both systems but their rapid movement prevented us from giving a precise along-track colocation except for the short scales close to the temporal crossing point. Indeed, for advective dynamics to be resolved correctly, they should conform to the Friedrichs-Lewy condition, ie $U \, \Delta t / \Delta x < 1$. If we are following small structures with typical advection speeds of U=0.3 m/s (typical of the Northern Current), then we need time differences, $\Delta t$, less than 1.35 days to resolve the smaller SARAL wavelengths at 35 km; and within 2 days for the Jason-2 and Cryosat-2 data resolving 50 km wavelength structures. With the slow-moving gliders, we can only cover 30 km per day, and so our along-track intercomparisons should be limited to the +/- 30 km around the altimeter-glider crossing point. This places a very strong constraint on our in-situ validation.

The Saral intercomparison with the Toulon HFradar data was quite promising. Despite the apparent nearshore errors in the Saral data, and the periods with strong wind-driven currents, the correlation between the Saral geostrophic currents and HFradar total currents remained high. The position of the Toulon HFradar helps, as the observations are centred on the Northern Current, in a region where the current is strongly steered by bathymetry, and the geostrophic component is dominant. This example indicates that a strong coastal current, with a high signal-to-noise, can be detected by satellite altimetry, even at 20 km from the coast. Improvements are still needed to reduce the altimetric errors in the nearshore region, and to compare the Cryosat-2 SAR current observations with the HFradar data. This good intercomparison suggests that





HFradar data may be combined with altimetry to extend the duration and offshore recirculation associated with the Northern Current near Toulon.

For the future altimetric missions, finer spatial sampling and lower noise levels should continue, with Sentinel-3 in global SAR mode launched in early 2016, and SWOT providing 2D interferometric SAR heights and images and an order of

5 magnitude lower noise in 2021. Similar wavenumber spectral analysis techniques could be applied to estimate the noise levels and observable spatial scales with these new missions. This study illustrates that the difficulties in setting up an adequate in-situ validation for the small-scale, rapidly evolving dynamics will remain a challenge to resolve in the future.

**Author contributions**: This work was carried out by A. Carret as part of her Masters program. R. Morrow supervised the

10 work and prepared the manuscript with contributions from all co-authors. G. Valladeau and F. Boy provided co-supervision. F. Birol and F. Nino provided support with the analysis. C. Bachelier processed the glider data and B Zakardjian the HFradar data.

**Competing Interests.** The authors declare that they have no conflict of interest.

15 **Acknowledgments.** This work was funded by an OSTST CNES TOSCA grant. The glider and HFradar data were funded as part of the French MOOSE Mediterranean observing system program, with additional finance from CNES as part of the Comsom glider campaign.



**Table 1. Altimetric data used in this study.**

| Altimetric Mission | Frequency Band | High-freq rate (average 1 Hz)[1] | Time period used | # sections used in spectral averages Mean (seasonal)[2] |
|---|---|---|---|---|
| **Jason-2** | Ku | 20 Hz - LRM | Jul 2008 – Feb 2015 | 246 (Sum: 65, Win: 58, Spr: 71, Aut: 52) |
| **SARAL** | Ka | 40 Hz - LRM | Mar 2013 – Jan 2015 | 292 (Sum: 66, Win: 66, Spr: 96, Aut: 64) |
| **Cryosat-2** | Ku | 20 hZ - SAR | Apr 2013 – Apr 2014 | 276 (Sum: 77, Win: 69, Spr: 75, Aut: 55) |

1. LRM : Conventional Low-resolution Mode; SAR : Synthetic Aperture Radar mode
2. First number corresponds to the total number of 200 km sections used in the regionally averaged spectra (Fig. 3); second numbers in brackets correspond to the number of sections used in each seasonal average (Fig. 4).

**Table 2. Characteristics of the collocated glider and altimeter track sections.**

| Altimeter Track | Alongtrack filtering[1] | Glider name | Start date of section | End date of section | Section length (km) | # glider profiles[2] |
|---|---|---|---|---|---|---|
| **Jason 146** | **50** | Campe | 23-Sep-2012 | 8-Oct-2012 | 292 | 111 |
| **Jason 146** | **50** | Campe | 8-Oct-2012 | 23-Oct-2012 | 327 | 80 |
| Saral 846 | **35** | Eudoxus | 23-Oct-2014 | 29-Oct-2014 | 125 | 54 |
| Saral 57 | **35** | Milou | 27-Oct-2014 | 3-Nov-2014 | 164 | 92 |
| Saral 388 | **30** | Milou | 9-Nov-2014 | 13-Nov-2014 | 77 | 55 |
| **Saral 973** | **35** | Bonplan | 13-Apr-2015 | 22-Apr-2015 | 180 | 101 |
| **Saral 973** | **35** | Tintin | 17-Apr-2015 | 23-Apr-2015 | 115 | 58 |
| **Saral 973** | **35** | Tintin | 8-May-2015 | 13-May-2015 | 99 | 56 |
| **Cryosat 493** | **35** | Bonplan | 24-Apr-2015 | 1-May-2015 | 166 | 101 |
| **Cryosat 493** | **35** | Tintin | 25-Apr-2015 | 4-May-2015 | 188 | 101 |

[1]Altimetric data are filtered with a Loess filter at different wavelength cutoffs depending on the mission and season (see text).

5  [2]All glider data are filtered with a 2-step Butterworth filter which removes high-frequency signals < 30 km wavelength.





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





**Figures**

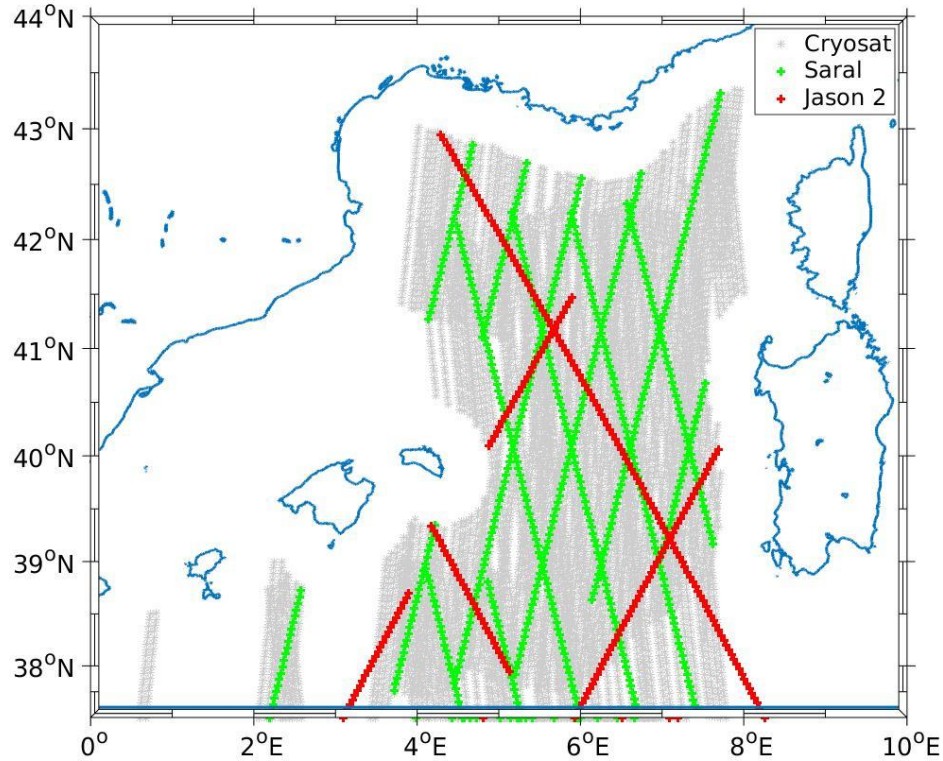

**Figure 1**. Distribution of altimeter tracks in the NW Mediterranean Sea showing the different missions: the 10-day repeat Jason-2 mission in red, 35-day repeat Saral/AltiKa in green, and the 380 day repeat Cryosat-2 in gray. Only sections greater than 200 km are included in the spectral analysis, and only data more than 50 km from the coast are analysed to remove the increased errors in the coastal zone. The distance from the coast is calculated using the Stumpf data base (http://oceancolor.gsfc.nasa.gov/DOCS/DistFromCoast)





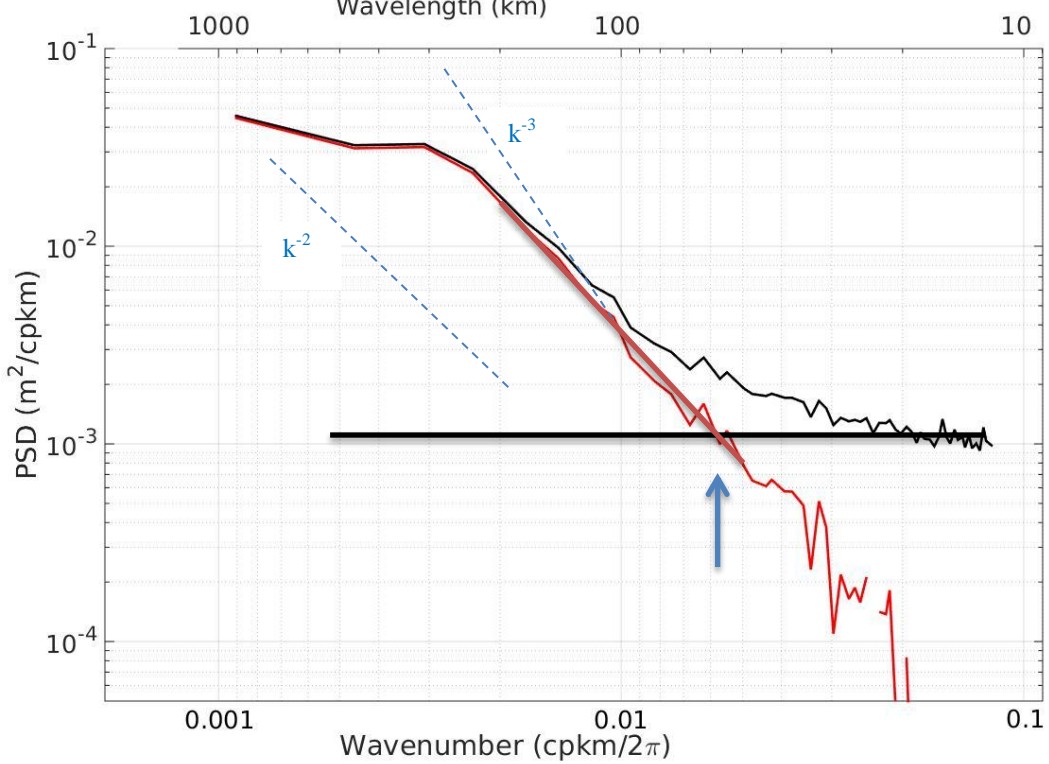

**Figure 2**. Mean wavenumber spectra (power spectral density) for Jason-2 sea level anomalies, averaged over all tracks in the NW Mediterranean Sea > 50 km from the coast (black line) for the period 1 Apr 2013 – 30 Apr 2014. The estimated noise level is shown as the thick black line. The unbiased spectra (thin red line) is obtained by subtracting this constant noise from the original spectra. The spectral slope (thick red line) is calculated between 50 and 200 km wavelength. The intersection between these two curves represents the mesoscale observational limit (blue arrow), above which the mean signal-to-noise ratio is >1.





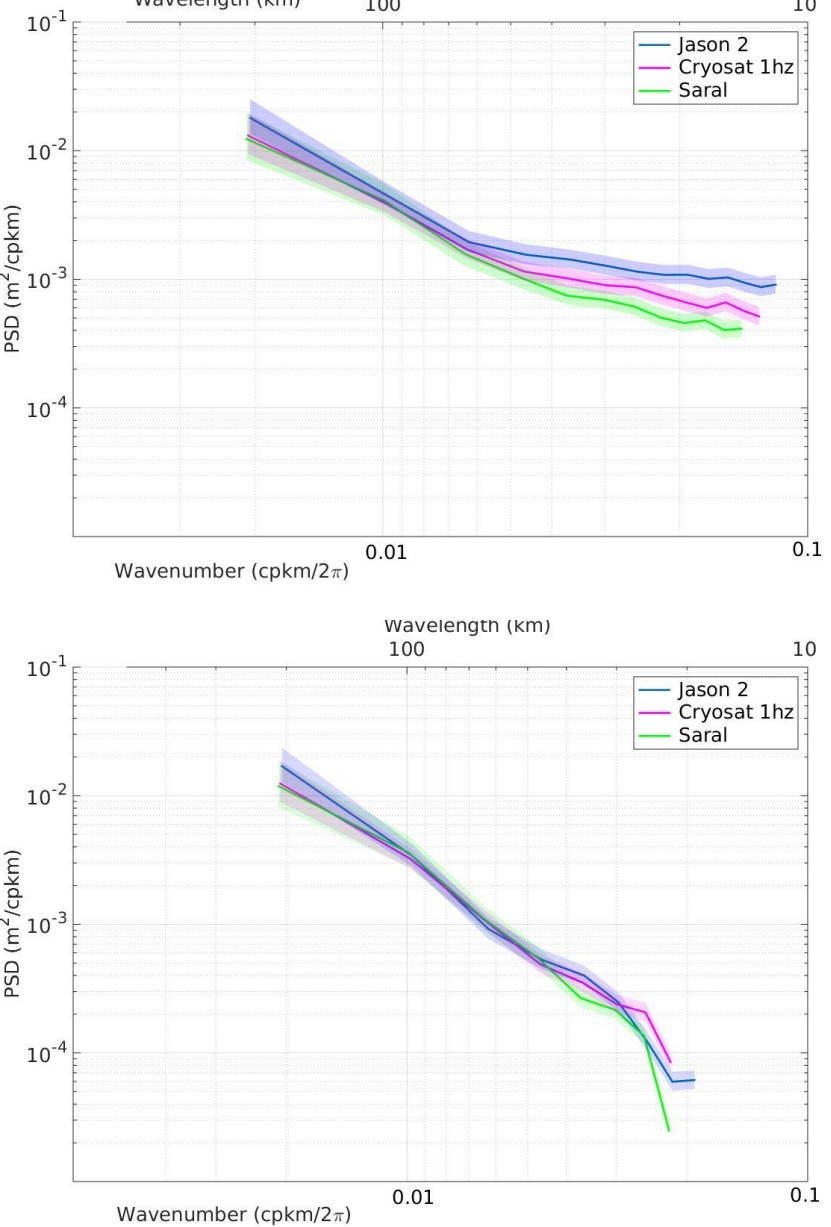

**Figure 3** a) Mean wavenumber spectra (power spectral density) for the three altimeter missions, averaged over all tracks in the NW
Mediterranean Sea > 50 km from the coast for the period 1 Apr 2013 – 30 Apr 2014. Jason-2 is in blue, Saral in green, Cryosat-2 SAR 1
hz data in pink. b) The unbiased spectra with a constant noise level removed, with a mean $k^{-2.5}$ spectral slope. Shading represents the error
bars, based on a Chi$^2$ test with the number of degrees of freedom being wavenumber dependant. Table 1 gives the number of sections used.




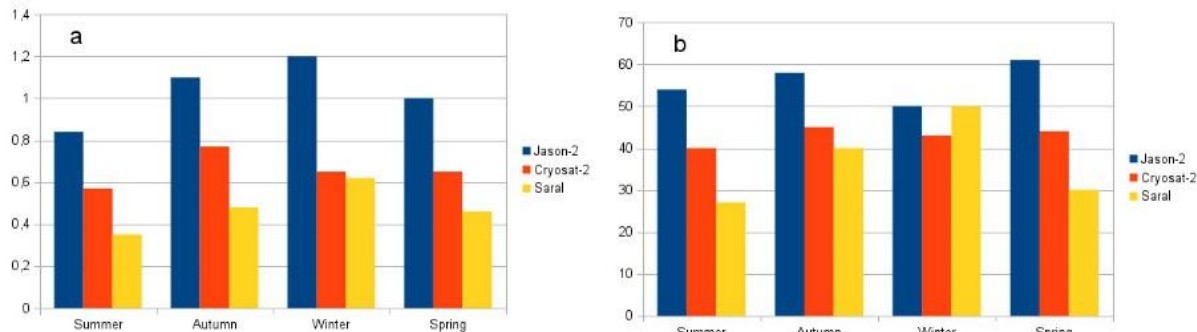

**Figure 4**. a) Seasonal noise levels (in $10^{-3}$ m$^2$/cpkm) for Jason-2 (blue), Cryosat-2 SAR mode (orange) and Saral/AltiKa (yellow) derived from along-track wavenumber spectra. b) Seasonal observational limits in terms of wavelength (in km) where the signal-to-noise is > 1 for each altimeter mission. Table 1 gives the number of sections used.



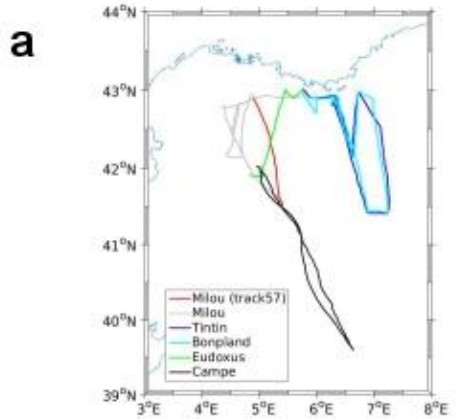

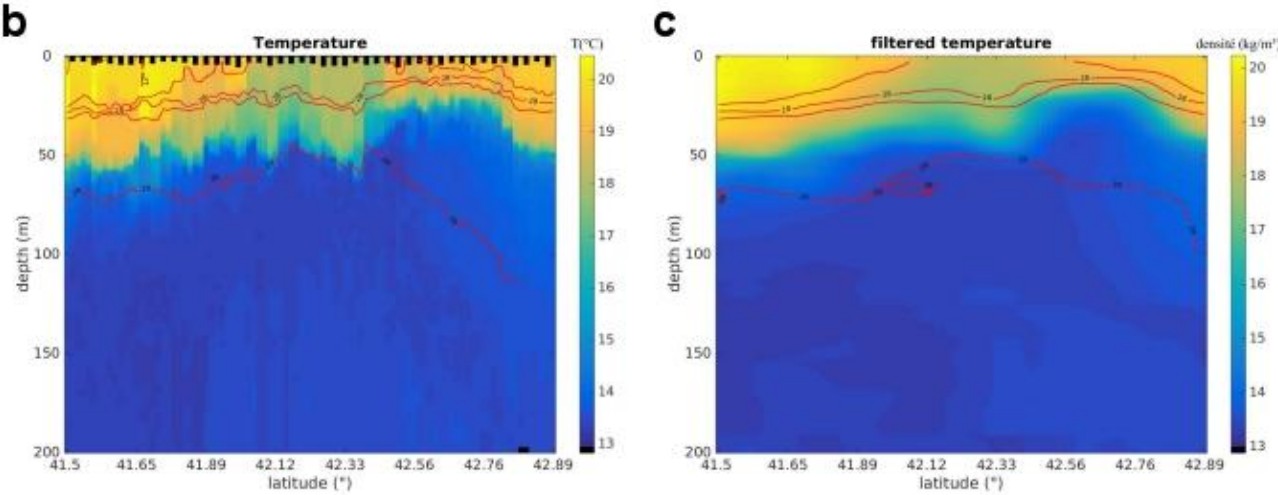

**Figure 5.** a) Location of the different gliders used in this analysis. In red, the glider "Milou" section (155 km long) along the Saral altimeter track 57 from the 27 Oct to 3 Nov 2014. b) Vertical temperature section from the Milou glider over the upper 200 m. c) Filtered temperature section with cutoff at 30 km wavelength.





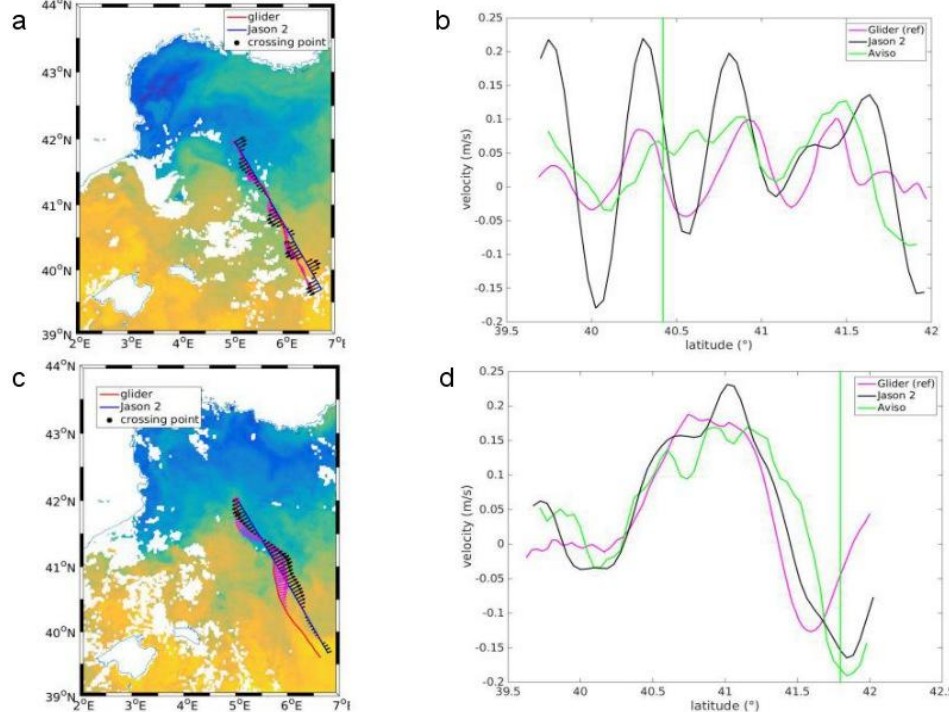

**Figure 6.** a) Collocated Jason-2 track and currents (in black) and glider track and currents (in pink) for the southbound leg, overlaid with a satellite SST plot on 1 Oct 2012. b) along-track comparison of geostrophic velocities for the glider (including the drift velocities) in pink, and filtered alongtrack Jason-2 data in black. Mapped AVISO altimeter data, interpolated back onto the Jason-2 track, are in green. Green vertical line shows the position when the Jason-2 data and gliders are collocated in time. c) and d), same but for the northbound section with SST fields from the 21 Oct 2012. 48h SST fields at 0.02° resolution from CLS.





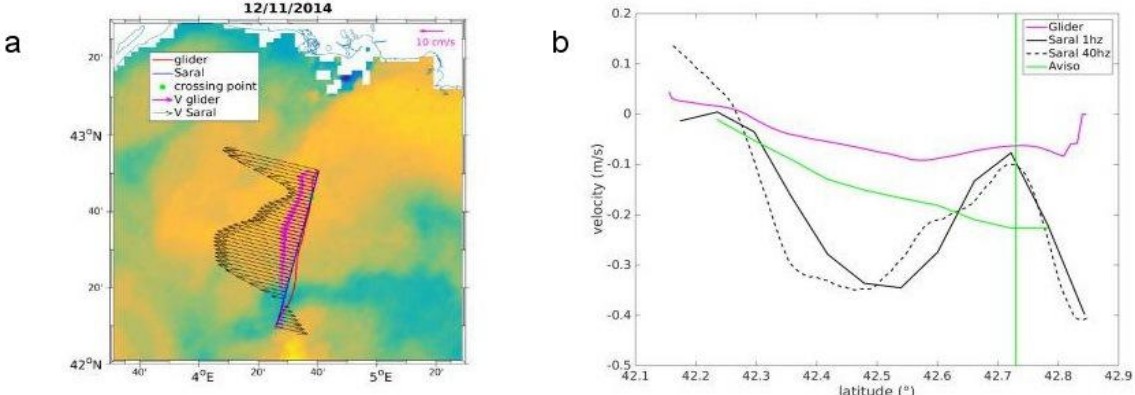

**Figure 7.** a) Collocated Saral track 388 and currents (in black) and glider track and currents (in pink), overlaid with a satellite SST plot on 12 Nov 2014. b) along-track comparison of geostrophic velocities for the glider (including the drift velocities) in pink, and filtered Saral data in black. Mapped AVISO altimeter data, interpolated back onto the altimeter track, are in green. Green vertical line shows the position when the altimeter data and gliders are collocated. Daily SST fields at 0.02° resolution from CLS.





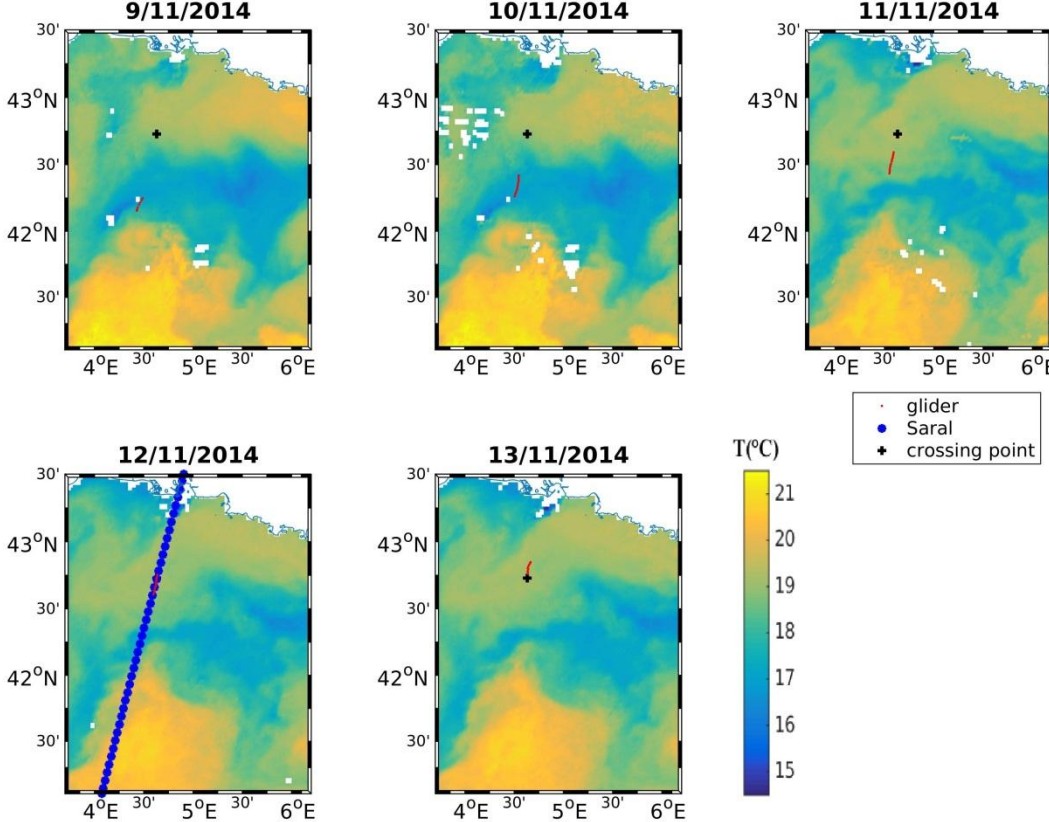

**Figure 8**. Five-day series of satellite SST maps for the period 9-13 November. The glider position is shown each day (in red), the Saral-glider crossing position on 12 November (in black), and the Saral track passing on 12 Nov 2014. Daily SST fields at 0.02° resolution from CLS.





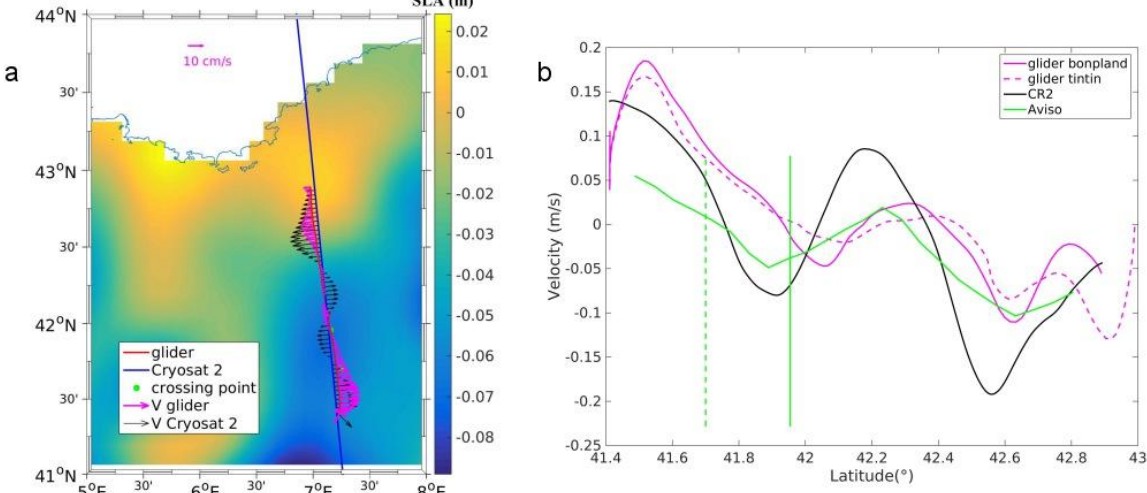

**Figure 9.** a) Collocated Cryosat-2 track 493 currents (in black) and glider currents (in pink), overlaid with a satellite SST plot on 27 Apr 2015. Two gliders, Bonpland and Tintin, follow at 1 day intervals. b) along-track comparison of geostrophic velocities for the Bonplanb glider (pink solid), and Tintin (pink dashed) with the filtered Cryosat-2 SAR data in black. Mapped AVISO altimeter data, interpolated back onto the altimeter track, are in green. Green vertical line shows the position when the altimeter data and gliders are collocated. (solid for Bonpland; dashed for Tintin). Daily SST fields at 0.02° resolution from CLS.



**Figure 10.** HF radar surface currents near Toulon for one date (20 Oct 2013); direction with small blue arrows, current speed is in colour. Saral track #302 is marked in pink. 1 Hz cross-track geostrophic currents from Saral altimetry are in black, the HF radar total currents projected in the altimetric cross-track direction are in red. The current scale of 30 cm/s is associated with the projected currents.





**Figure 11** a) Upper panel: 18 month time series of daily HF radar surface currents projected in the cross-track direction of the Saral groundtrack. Red contours at -0.3 m/s aid to delimit the westward Northern Current position. B) Middle panel: Extraction of these daily HF radar currents at the day of the Saral observations. The temporal mean value is shown on the left. C) Bottom panel: cross-track geostrophic currents from the Saral altimeter data, filtered at 45 km wavelength. Arrows mark the dates with low correlations < 0.5.



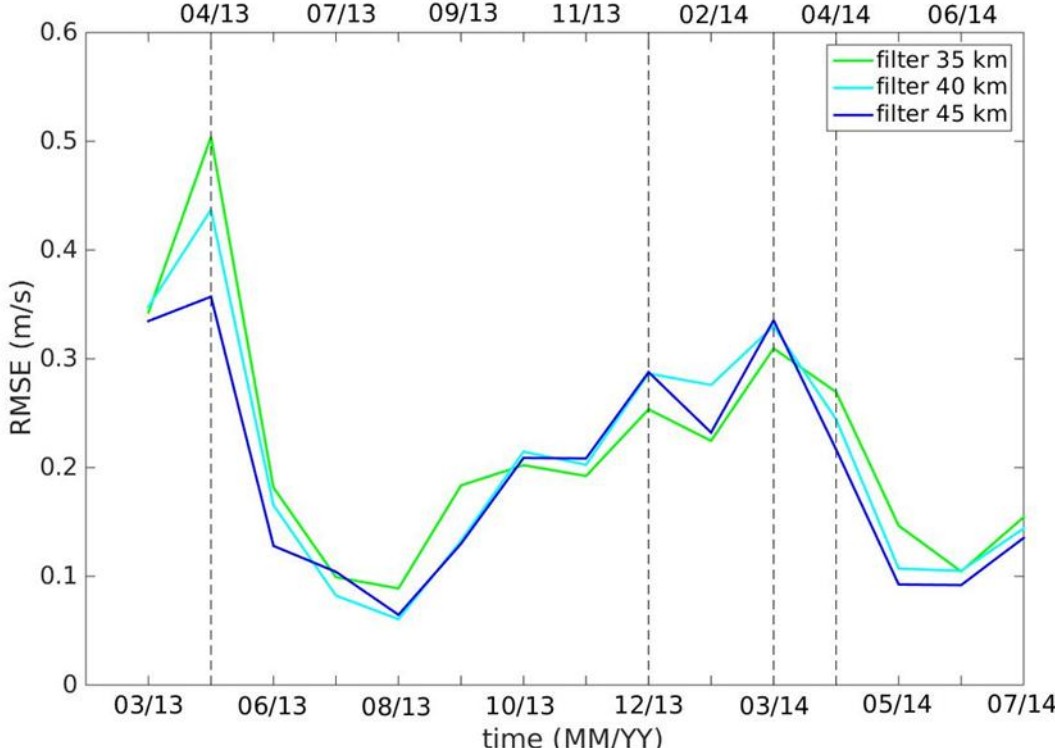

**Figure 12** RMSE between the cross-track HF radar current amplitudes and the Saral current amplitudes. Dates with low correlations (<0.5) are marked with the vertical dashed line.