# Peer review of "Observability of fine-scale ocean dynamics in the Northwest Mediterranean Sea"

_Ocean Science, 2016_

## Referee Comment (RC1) · Anonymous Referee #1 · 2 Nov 2016

This paper investigates the noise levels for recent altimeter missions (Jason-2, CryoSat-2 and AltiKA), as emerging from the computation of along-track spectra. This technique has been already applied to global altimeter data sets, for Jason-1 (Xu and Fu 2012) and for Jason-2, AltiKA and CryoSat-2 (Dufau et al. (2016). The authors replicate the analysis in the NW Mediterranea Sea that is an excellent laboratory to assess finer-scale ocean processes with along-track altimeter data.

The authors use 1 Hz data that have a flat noise floor – the higher frequency (20 Hz or 40 Hz data) showing a spectral bump at wavelengths less than 70 km which does not allow us to estimate a stable noise floor. The authors show how much of the ocean dynamical signal is observable above the noise. The smallest scales observable vary from one altimeter mission to another and seasonally. In the study area, the results show that we cannot observe structures less than 35-45 km wavelength with

AltiKA, 50-60 km wavelength with the higher noise of Jason-2 and CryoSat-2 blocks the observation of scales less than 50-55 km.

As the point is that energetic features can be observable above statisitcal noise, the authors present and discuss specific case-studies where ground-truth is available at time of altimeter passages. One case-study is a series of collocated along-track altimeter-glider sections where geostrophic velocities derived from the two independent data sets are compared. The second case-study utilizes co-located HF radar to compare the oceanic surface currents.

This paper contributes to better understanding the potential of along-track altimeter data in observing the ocean variability at small scale, considering that improved capabilities of new satellite altimetry missions and refined processing are expected to provide more and better data than past. The paper shows evidence that along-track filtering usually applied to reduce the instrument and geophysical noise should not be applied in a similar way to all altimeter missions but also adapted to the regional conditions. The results also show the interpretation difficulties encountered in comparing with in-situ data at short scale, especially when there is a rapidly evolving ocean dynamics and along-track altimetry data are not exactly and collocated in time and space with in situ observations.

Overall, the paper is well organized and easy to read with very good English. I recommend publication after minor revision. I only strongly suggest authors to include and consider an analysis at cross-over points, as CryoSat-2 due to its non-geodetic orbit provides lot of crosses, some probably near coincident in time with the other satellites over the 13-month common data period from 1 April 2013 to 30 April 2014.

Additional minor comments:

Pg 3, Row 10, "seasons.," – typo to be corrected Pg 3, Row 25, "..The Mediterranean Sea, dominated by small dynamical structures, may have different spectral energy and spectral slopes than in other open ocean regions.." – this statement is not proved; it

seems just a speculation Pg 6, row 21, "SSH PSD" – somewhere you state SLA and now SSH. is PSD computed using SSH or SSHA (anomalies) ? Pg 13, row 29 and row 34, "HFradar" – separate HF from radar Pg 14, row 1, "HFradar" – separate HF from radar

---

## Referee Comment (RC2) · A. Sánchez Román (Referee) · 11 Nov 2016

In this work, the authors investigate both the noise level and observable ocean scales for three altimeter missions (Jason-2, Saral/AltiKa and Cryosat-2) in the northwestern Mediterranean Sea through a spectral analysis approach. Along-track satellite data collected over several years are used. Furthermore, comparisons with in-situ observations from glider and HF radar in different individual case studies are also performed in order to validate the altimetric data.

I think that this paper presents original work and I consider that it must be published in Ocean Science with some minor corrections. I have, however, some suggestions and questions that should be taken into account and reasonably solved. There are also some typographical and grammatical errors that need to be corrected.

[Figure]

In the following there is the list of my comments:

[1] My first comment is related to the filters applied to both altimetry and glider data. In Table 2, authors indicate that altimetric data is filtered by applying a Loess filter while glider data is filtered by using a 2-step Butterworth filter. Why the authors did not use the same kind of filter for both datasets? Is the choice of the filters based on previous studies? I think that this issue should be clarified and an explanation should be included in the text. On the other hand, it is not clear to me how the authors have chosen the filtering scales showed in Table 2 for the different altimeter missions. Did they perform a sensitivity test in order to choice the more suitable window cut-off or it is based on the spectral analysis results? If the latter applies and that is also the reason for choosing a filtering scale of 30 km in Saral track #388, it should be specified in the text to avoid confusion.

[2] In the first sentence of section 2.3 (line 25 of page 5) the meaning of "over a number of year" is not clear. I guess that authors refer to the time period over which the radar has been working. If so, please indicate this time period.

[3] in line 31 onwards of the same page it sounds better "The system uses two WERA radars that provide surface current vectors over a region extending 80-100 km offshore, with a spatial resolution of 3 km and an angular resolution of 2 degrees. They operate at 16-17 Mhz. Observations are collected every 20 min and data have been edited and averaged . . ."

[4] In section 3 (line 6 of page 6), it should be written "than 50 km from the coast are analysed to avoid the increased errors in the coastal zone" because the authors did not apply any procedure to remove these errors, but selected a dataset with a typical reduced coastal noise.

[5] Concerning to the previous point, why authors selected segments of 200 km? Please give a reasonable explanation.

[6] In lines 11 - 13 of page 6 (section 3), authors state that "an example of the power spectral density (PSD) of sea level anomaly averaged for all of the Jason-2 data in the NW Mediterranean Sea is shown as the black curve in Fig. 2". This is not correct because in this Figure the dataset used spans from 1st April 2013 to 30th April 2014 while the whole dataset investigated for the Jason-2 data (Table 1) spans from 2008 to 2015. Actually, this is the common period investigated in the three satellite missions (given in line 28 of the same page) for the spectral analysis. Therefore, this sentence should be reworded to properly indicate the time-period used.

[7] The red line in Figure 2 showing the spectral slope is not easily observed. Please change its color. Moreover, in line 17 of section 3 it should be written "black line" instead of "red line" according to plots in Figure 2. Also, in line 19 it should appear "red line" instead of "black line"

[8] As an overall comment, authors should be consistent with the dimension of the units along the text. Sometimes velocities are given in m/s and sometimes in cm/s. Since units in all figures are given in m/s, I strongly recommend putting all velocities in m/s within the text. Furthermore, in section 4 velocities and times are expressed in km/sec, secs and m/sec. It is more appropriate to refer time as "s" instead of "sec".

[9] Label of color bar in Figure 5.c indicates density (kg/m3). Please change to T ($^\circ$C) [10] In line 9 of page 9 (section 4.1) it should be written "(in pink)" instead of "(in red)" in order to be consistent with Figure 6.

[11] In line 11 of page 10 it sounds better "Figure 8 shows the five days needed by the glider to . . ." Please change it in the text.

[12] In the first sentence of section 5 (page 11) please remove "an" since "additional means" is plural. Furthermore, in line 15 change "leaving the geostrophic . . ." by "retaining the geostrophic . . ." this is more formal.

[13] In line 22 of page 12 please remove the word "altimetric" since it is redundant.

[14] Red arrow denoting the current scale in Figure 10 should be in m/s in order to be consistent with velocities displayed in the figure. The same applies to panel a in Figure 7. Moreover, label of color bar in Figure 10 should be "m/s" or "m s -1" instead of (m/s-2).

[15] Finally, caption of Figure 11 indicates that Saral data has been filtered at 45 km wavelength but according both to the text and Table 2 it is filtered at 35 km. Please change it.

---

## Author Comment (AC1) · 9 Dec 2016

Firstly, we wish to thank the reviewer for providing interesting and constructive comments to this paper.

Detailed response to Reviewer 1's comments :

Reviewer comment : I only strongly suggest authors to include and consider an analysis at cross-over points, as CryoSat-2 due to its non-geodetic orbit provides lot of crosses, some probably near coincident in time with the other satellites over the 13-month common data period from 1 April 2013 to 30 April 2014.

Reply : This is a very good point. There are potential crossover points during this period from Cryosat-2 on its long-repeat 369-day orbit and even from Jason-1 which moved

into a long-repeat 406-day geodetic orbit from April 2012-1 July 2013. Our analyses of the small, fast-moving features in this paper indicated that we really need crossover measurements overlapping within 1-2 days to capture these fine-scale features. These multi-altimeter overlapping passes are also interesting for the missions on a similar inclination, since their overlapping sections can be quite long, eg Saral & Cryosat may have long overlapping sections with a time difference of less than 2 days. Similar long sections may be available from the Jason-1 geodetic mission & Jason-2. At present, we are developing the code to calculate the crossovers from multi-satellite passes and select the passes based on their time differences. This analysis is not available yet, and will not be included in the present paper, but will be continued as part of the PhD work of Alice Carret. A note on this is now included in the discussion.

Additional minor comments:

Pg 3, Row 10, "seasons,." – typo to be corrected Reply : Corrected

Pg 3, Row 25, "..The Mediterranean Sea, dominated by small dynamical structures, may have different spectral energy and spectral slopes than in other open ocean regions.." – this statement is not proved; it seems just a speculation Reply : Indeed this was not proven here. However, the arguments behind this sentence were to argue about the effects of calculating spectral slopes over a fixed "mesoscale" wavelength band over the global oceans, and the impact of this fixed wavelength band for the Mediterranean Sea where the Rossby radius is quite small. This sentence has been modified to include a clearer discussion on this key point, as follows : "These studies calculated their spectral slopes over a fixed "mesoscale" band from 70-250 km wavelength. The Mediterranean Sea, which is dominated by smaller dynamical structures, may have different spectral energy and spectral slopes in this band compared to open ocean regions."

Pg 6, row 21, "SSH PSD" – somewhere you state SLA and now SSH. is PSD computed using SSH or SSHA (anomalies) ? Reply : We apologise for this confusion. We have

added a sentence in the data processing section (end of section 2.1) to clarify that we use the SLA in our analyses. SSH has been replaced by SLA in the rest of the paper. "In the following analyses of spectra and geostrophic current anomalies, we will use the time-varying SLAs."

Pg 13, row 29 and row 34, "HFradar" – separate HF from radar Pg 14, row 1, "HFradar" – separate HF from radar Reply : Done

---

## Author Comment (AC2) · 9 Dec 2016

Detailed response to Reviewer 2's comments : [1] My first comment is related to the filters applied to both altimetry and glider data. In Table 2, authors indicate that altimetric data is filtered by applying a Loess filter while glider data is filtered by using a 2-step Butterworth filter. Why the authors did not use the same kind of filter for both datasets? Is the choice of the filters based on previous studies? I think that this issue should be clarified and an explanation should be included in the text. On the other hand, it is not clear to me how the authors have chosen the filtering scales showed in Table 2 for the different altimeter missions. Did they perform a sensitivity test in order to choice the more suitable window cut-off or it is based on the spectral analysis results? If the latter applies and that is also the reason for choosing a filtering scale of 30 km in Saral track #388, it should be specified in the text to avoid confusion.

[Figure]

Reply : The filtering for the alongtrack altimetry data is based on the standard filtering applied to the CTOH coastal processed data (Birol et al., 2010; Birol and Nino, 2015). The glider filtering is based on the 2-step Butterworth filter to remove the high-frequency noise below the Rossby radius, based on expertise with Saral AltiKa data where numerous filtres were tested. The references for these 2 choices are now more clearly expressed in the paper. For the final filtering scales applied in Table 2, the first estimate was based on the seasonal spectral analysis results. However other cut-off frequencies were tested. The filter which gave the best results in terms of correlation coefficient and which had the lowest cut-off wavelength was then chosen. This detailed description has also been added to the text in the introduction to section 4 (glider-altimeter comparisons).

[2] In the first sentence of section 2.3 (line 25 of page 5) the meaning of "over a number of year" is not clear. I guess that authors refer to the time period over which the radar has been working. If so, please indicate this time period.

Reply : This has been replaced by "with gridded data available since 2012"

[3] in line 31 onwards of the same page it sounds better "The system uses two WERA radars that provide surface current vectors over a region extending 80-100 km offshore, with a spatial resolution of 3 km and an angular resolution of 2 degrees. They operate at 16-17 Mhz. Observations are collected every 20 min and data have been edited and averaged:::"

Reply : This has been re-worded accordingly

[4] In section 3 (line 6 of page 6), it should be written "than 50 km from the coast are analysed to avoid the increased errors in the coastal zone" because the authors did not apply any procedure to remove these errors, but selected a dataset with a typical reduced coastal noise.

Reply : OK. This has been reworded as : "This segment length was chosen to allow

a large number of altimeter segments in different regions in between the numerous islands and to be more than 50 km from the coast, to avoid the increased errors in the coastal altimeter data."

[5] Concerning to the previous point, why authors selected segments of 200 km? Please give a reasonable explanation.

Reply : This has been reworded – see previous comment [4] [6] In lines 11 - 13 of page 6 (section 3), authors state that "an example of the power spectral density (PSD) of sea level anomaly averaged for all of the Jason-2 data in the NW Mediterranean Sea is shown as the black curve in Fig. 2". This is not correct because in this Figure the dataset used spans from 1st April 2013 to 30th April 2014 while the whole dataset investigated for the Jason-2 data (Table 1) spans from 2008 to 2015. Actually, this is the common period investigated in the three satellite missions (given in line 28 of the same page) for the spectral analysis. Therefore, this sentence should be reworded to properly indicate the time-period used.

Reply : this has been reworded to specify that the example in Figure 2 covers the longer time period from 2008 to 2015

[7] The red line in Figure 2 showing the spectral slope is not easily observed. Please change its color. Moreover, in line 17 of section 3 it should be written "black line" instead of "red line" according to plots in Figure 2. Also, in line 19 it should appear "red line" instead of "black line"

Reply : The red slope line has been replaced by a red dashed line to be clearer. The text has been modified for these points to specify the correct line color in the text and in the Figure. We apologise for this.

[8] As an overall comment, authors should be consistent with the dimension of the units along the text. Sometimes velocities are given in m/s and sometimes in cm/s. Since units in all figures are given in m/s, I strongly recommend putting all velocities in m/s

within the text. Furthermore, in section 4 velocities and times are expressed in km/sec, secs and m/sec. It is more appropriate to refer time as "s" instead of "sec".

Reply : this has been changed throughout.

[9] Label of color bar in Figure 5.c indicates density (kg/m3). Please change to T (C)

Reply : this has been modified

[10] In line 9 of page 9 (section 4.1) it should be written "(in pink)" instead of "(in red)" in order to be consistent with Figure 6.

Reply : The Figure has been modified to be in pink, to be consistent with Figure 6b as well. The text remains "in pink".

[11] In line 11 of page 10 it sounds better "Figure 8 shows the five days needed by the glider to :::" Please change it in the text.

Reply : Done

[12] In the first sentence of section 5 (page 11) please remove "an" since "additional means" is plural. Furthermore, in line 15 change "leaving the geostrophic ::: " by "retaining the geostrophic:::" this is more formal.

Reply : Done

[13] In line 22 of page 12 please remove the word "altimetric" since it is redundant.

Reply : Done

[14] Red arrow denoting the current scale in Figure 10 should be in m/s in order to be consistent with velocities displayed in the figure. The same applies to panel a in Figure 7. Moreover, label of color bar in Figure 10 should be "m/s" or "m s 1" instead of (m/s-2).

Reply : Done

[15] Finally, caption of Figure 11 indicates that Saral data has been filtered at 45 km wavelength but according both to the text and Table 2 it is filtered at 35 km. Please change it.

Reply : Done